# Photovoltaic Device Application of a Hydroquinone-Modified Conductive Polymer and Dual-Functional Molecular Si Surface Passivation Technology

**DOI:** 10.3390/polym14030478

**Published:** 2022-01-25

**Authors:** Na Yeon Park, Gwan Seung Jeong, Young-Jin Yu, Yoon-Chae Jung, Jin Hee Lee, Jung Hwa Seo, Jea-Young Choi

**Affiliations:** 1Department of Metallurgical Engineering, Dong-A University, Busan 604-714, Korea; nayeon2385@gmail.com (N.Y.P.); wjdrhkstmd12@gmail.com (G.S.J.); tq6284@naver.com (Y.-J.Y.); dbsco0306@naver.com (Y.-C.J.); 2Department of Chemical Engineering, Dong-A University, Busan 604-714, Korea; dljh82@gmail.com (J.H.L.); seojh@dau.ac.kr (J.H.S.); 3Department of Materials Sciences & Engineering, Dong-A University, Busan 604-714, Korea

**Keywords:** PEDOT:PSS, heterojunction solar cell, work-function, conductivity, passivation

## Abstract

In the last decades, the conductive polymer PEDOT:PSS has been introduced in Si-based hybrid solar cells, gaining noticeable research interest and being considered a promising candidate for next generation solar cells which can achieve both of low manufacturing cost and high power conversion efficiency. This study succeeded in improving the electrical conductivity of PEDOT:PSS to 937 S/cm through a simple process of adding hydroquinone (HQ) to the pristine PEDOT:PSS solution. The results also showed that the addition of HQ to PEDOT:PSS(HQ-PEDOT:PSS) could not only dramatically improve the conductivity but also well-sustain the work function characteristics of PEDOT:PSS by promoting the formation of more continuous conductive-PEDOT channels without removing the insulating PSS. In this report, we reveal that the application of the HQ-PEDOT:PSS to the Si/PEDOT:PSS HSC could significantly improve the short-circuit current and open-circuit voltage characteristics to increase the power conversion efficiency of the HSCs compared to the conventional approaches. Moreover, we also treated the Si surface with the organic monomer, benzoquinone (BQ) to (1) passivate the excess Si surface defect states and (2) to improve the properties of the Si/PEDOT:PSS interface. We show that BQ treatment is able to dramatically increase the minority carrier lifetime induced by effective chemical and field-effect passivation in addition to enhancing the wettability of the Si surface with the PEDOT:PSS solution. As a result, the power conversion efficiency was increased by 10.6% by introducing HQ and BQ into the fabrication process of the Si/PEDOT:PSS HSC.

## 1. Introduction

Photovoltaic (PV) cells have become increasingly important in their role as harvesters of sunlight as a major source of sustainable and renewable energy. Research devoted to the use of PVs to solve global environmental problems arising from growing energy shortages and greenhouse gas emissions has been actively ongoing for the past few decades. [1,2,3], Currently, monocrystalline and polycrystalline silicon (Si)-based PV technologies account for around 90% of the total solar cell market because of their non-toxicity, well established advanced microelectronics manufacturing technology, and long-term stability [4,5,6,7]. Conventional Si solar cells have a p-n junction structure which is known to provide highly effective charge separation and collection of photogenerated carriers in devices. Generally, the p-n junction structure is formed by using a dopant diffusion process which requires high-temperature conditions (≥800 °C), which inevitably complicate the process and increase the manufacturing cost [8,9,10], From this standpoint, Si/organic heterojunction solar cells (HSCs) offer advantages compared with conventional cells in that Si/organic HSCs can be fabricated using low-temperature (≤200 °C) and low-cost processes (e.g., spin-coating) and offer a simple device structure. These advantages have resulted in HSCs being widely studied as the next-generation PV technology [11,12,13]. HSC devices are generally fabricated by using organic materials for the carrier selective layer to promote the extraction efficiency of photogenerated carriers in the devices. A wide range of organic materials, such as conductive polymers [14,15] small molecules [16,17] and fullerene derivatives [18,19] are currently being studied for this application.

Among all these materials, the *p*-type conductive polymer, poly(3,4-ethylenedioxythiophene):poly(4-styrenesulfonate) (PEDOT:PSS), is one of the most widely studied materials for inclusion as the hole selective/transport layer or the transparent electrode material when fabricating HSCs owing to its relatively high thermal stability, electrical conductivity, and ability to transmit light [20,21,22,23,24]. However, coated PEDOT:PSS thin films (TFs) have low electrical conductivity of less than 1 S/cm, and this has been recognized as one of the major barriers in the way of further increasing the power conversion efficiency (PCE) of HSCs based on Si/PEDOT:PSS [25]. This problem arises because PEDOT:PSS exists in the form of a core-shell structure (Figure 1a) composed of a PEDOT-rich core and a PSS-rich shell, owing to the large difference in their molecular weights, as schematically illustrated in Figure 1a [26,27,28]. As a result, the discontinuous distribution of PEDOT as the conductive component prevents efficient charge carrier transport in the TF, as illustrated in Figure 1b [29,30,31]. Moreover, the excess amount of PSS in the PEDOT:PSS solution (e.g., PEDOT:PSS = 1:2.5 for PH1000) initially results in the formation of a PSS-rich top layer on the surface during TF formation (Figure 1b), causing a further increase in the electric resistance of the film, which acts as a barrier for external charge carrier extraction [31,32].

Previous studies have reported an improvement in the electrical properties of PEDOT:PSS by using a secondary doping method to add polar solvents with a high boiling point, such as dimethyl sulfoxide (DMSO), glycerol, and ethylene glycol (EG). These solvents weaken the ionic interaction between PEDOT and PSS and then increase the size of the domains occupied by the conductive PEDOT in the PEDOT:PSS TFs [24,33]. Moreover, in addition to the above, the electrical conductivity can reportedly be improved by subjecting the coated PEDOT:PSS to post-surface treatment using DMSO, EG, or an inorganic acid as the solvent [24,30,34]. The purpose of this post-surface treatment is to remove the insulator-like PSS from the PSS-rich top layer that contains an excess of PSS [35,36]. However, conventional technologies not only involve a complicated process but also inevitably lower the work function (W.F.) of PEDOT:PSS when attempts are made to increase σ because they include a process to remove the insulator-like PSS which is initially responsible for the outstanding W.F. properties of PEDOT:PSS, namely ~5.0 eV [37,38]. The lower W.F. is considered to be one of the reasons for the decrease in the PCE of fabricated HSCs because it gives rise to a large reverse saturation current density (J_0_) and low open-circuit voltage (V_oc_) in the fabricated devices [39,40,41].

Moreover, as additional considerations for fabricating high-efficiency Si/PEDOT:PSS HSCs, it is also essential to form an effective Si surface passivation layer with hydrophilic properties. A layer such as this offers both an improved minority carrier lifetime (τ_eff_) by lowering the defect density of the Si surface and ensures intimate contact between the spin-coated PEDOT:PSS and Si. The most widely used hydrophilic passivation layer for Si is a silicon oxide (SiO_x_), which can be either naturally grown [42] or chemically grown using a boiling HNO_3_ solution [43], dry UV/O_3_ treatment, and ozonized deionized water [44]. This SiO_x_ layer improves the hydrophilicity and serves as a passivation layer for unsaturated Si bonds on the Si surface and thus enhances the properties of the interface of the Si/PEDOT:PSS HSC [43,45]. However, because effective external carrier extraction of the photo-generated carriers is a highly important pre-requisite for the efficiency of solar cells, the thickness of the SiO_x_ layer must remain less than 2 nm [46,47,48]. However, such a thin SiO_x_ layer would provide limited passivation effect for Si surface defect states and is the origin of the short τ_eff_ of photo-generated carriers in devices [49].

These shortcomings have motivated recent studies on Si surface passivation technology in which organic materials are actively being sought as replacements for conventional SiO_x_. Passivation of the Si surface with organic materials is particularly advantageous because it would enable τ_eff_ to be extended even with simple low-temperature and low-cost passivation processes. Previous studies for Si surface passivation with organic materials such as quinhydrone [50], 9,10-phenanthrenequinone [51], 1-octadecene [52], polyethyleneimine (PEI) [53], polyvinyl alcohol (PVA), and poly(methyl methacrylate) (PMMA) [54] succeeded in significantly improving τ_eff_ by forming effective chemical bonds with the unsaturated Si atoms present on the Si surface.

The objective of this study was to improve the PCE of Si/PEDOT:PSS HSC by applying our novel technologies to improve the conductivity of PEDOT:PSS and to effectively passivate the Si surface with the organic monomer benzoquinone (BQ). As reported, we successfully improved the conductivity of PEDOT:PSS to the level of 937 S/cm simply by mixing hydroquinone (HQ), which acts as a proton donating agent, with the pristine PEDOT:PSS solution. Our approach obviated the need for the PSS-removal process, indicating that it would also eliminate the problem of lowering the W.F. associated with conventional methods. Therefore, utilization of our HQ method for HSC fabrication was expected to improve the short-circuit current (J_sc_) and open-circuit voltage (V_oc_) of the devices. Moreover, in an attempt to further increase the PCE, benzoquinone (BQ) was introduced for the effective surface passivation of Si. In addition, the quality of the interface between Si and PEDOT:PSS, which would be induced by improving the wettability of PEDOT:PSS solution on the Si surface, would be enhanced. As a result, we successfully increased the PCE of the HSC to 10.6%, an improvement of more than 30% compared to the PCE attained with conventional technology.

## 2. Experimental Section

### 2.1. Preparation of PEDOT:PSS Solution

The PEDOT:PSS (HQ-PEDOT:PSS) solution containing HQ was prepared as follows: pristine PEDOT:PSS (Clevious PH1000, Heraeus, Hanau, Germany) was filtered with a PTFE syringe filter (JET BIOFIL, 0.45 μm), after which the solution was stirred overnight. Then, 3 mL of the prepared pristine PEDOT:PSS solution was mixed with various amounts of HQ (99%, Sigma Aldrich Burlington, MA, USA): 0.33 wt%, 0.67 wt%, and 1.0 wt%, respectively. The PEDOT:PSS (DMSO-PEDOT:PSS) solution containing DMSO was prepared by mixing 0.5 wt% of dimethyl sulfoxide (DMSO, JUNSEI, Tokyo, Japan) and 0.1 wt% of Triton X-100 (TX, DAEJUNG, Siheung, Korea) with pristine PEDOT:PSS as reported previously [55].

### 2.2. Device Fabrication

Double-side polished *n*-type Si (*n*-Si) wafers with a resistivity of 1.7–2.3 Ωcm, (100) orientation, and a thickness of 280 μm were cut into pieces sized 2 cm × 2 cm and cleaned with the aid of ultra-sonication in acetone, methyl alcohol, and distilled-water (DI-water), in that order, for 15 min each. Afterward, the wafers were additionally cleaned in an RCA solution containing a mixture of Na_4_OH, H_2_O_2_, and DI-water in a volume ratio of 1:1:5 at a temperature of 80 °C for 15 min to remove residual organic contaminants. Subsequently, the wafers were immersed in a dilute HF solution (1 vol%) for 1 min to remove the unintended native oxide layer from the Si surface. To fabricate the HSC with the SiO_x_ passivation layer, the HF-treated Si substrates were exposed to air at room temperature for 1 h, as reported in the previous study [43]. The BQ passivation solution was prepared by dissolving 10 mM benzoquinone (BQ, 99% ACROS) in 500 mL of methyl alcohol (MeOH, 99.5%, SAMCHUN Co., Seoul, Korea). Si surface passivation with BQ was accomplished by placing the HF-cleaned Si substrates in a jig that is open on the one side and then immersed in a BQ/MeOH solution at room temperature for 1 h to form the BQ passivation layer, following which the BQ-treated Si substrate was dried in a stream of N_2_. The prepared PEDOT:PSS solutions were spin-coated onto the Si surfaces on which the SiO_x_ and BQ passivation layers had been formed at 4000 rpm for 60 s followed by annealing at 170 °C for 10 min. Then, using thermal evaporation, the metal electrodes were deposited under vacuum at 3.0 × 10^−6^ mbar. The back electrode consisted of Al (200 nm)/LiF (1 nm) and a shadow mask was used to deposit a 200-nm-thick Ag grid to serve as the front electrode.

### 2.3. Characterization

The average sheet resistance value of the PEDOT:PSS TF was calculated after taking measurements at eight different locations per sample using a four-point probe system (CMT-SR2000N, AIT Co., Ltd., Suwon, Korea). The thickness of the TFs was measured with a surface profilometer (Tencor P6, KLA, Milpitas, CA, USA) and was used to derive the electrical conductivity of the coated TFs. The surface morphology of the PEDOT:PSS TF was studied by capturing atomic force microscopy (AFM, MultiMode-V, Veeco, Plainview, NY, USA) images in the tapping mode. The PSS-to-PEDOT ratio of coated PEDOT:PSS were measured by X-ray photoelectron spectroscopy (XPS) in the range of 157–175 eV and the W.F. of each TF was obtained with ultra-violet photoelectron spectroscopy (UPS, ESCALAB 250XI, Thermo-Fisher Co., Waltham, ,MA, USA). The wettability of the PEDOT:PSS solution on the Si surface was measured with a contact angle measurement system (FM-40, KRUSS Ltd., Hamburg, Germany). The minority carrier lifetime of the samples was measured with a photo-conductance lifetime tester (WCT-120, Sinton Instrument, Boulder, CO, USA), and the PCE of the fabricated HSCs was evaluated under the simulated air mass (AM) 1.5 G condition.

## 3. Result & Discussion

### 3.1. Fabrication of Si/PEDOT:PSS Heterojunction Solar Cells (HSCs)

This report introduces our novel approach to fabricate Si/PEDOT:PSS HSCs that utilize the proton-donating agent, HQ, to increase the conductivity (σ) of PEDOT:PSS [56]. Based on this report, various amounts of HQ were mixed with the pristine PEDOT:PSS solution to investigate the optimal amount of HQ that would most improve the performance of the HSCs. Before fabricating the HSCs, the σ of each of the HQ-PEDOT:PSS TFs was measured to observe the variation in σ with the amount of HQ, as shown in Figure 2a. As shown in the figure, 0.67 wt% of HQ resulted in the highest improvement in σ of 936.8 S/cm. This improvement is attributed to the added HQ acting as an H^+^ donor for PEDOT:PSS to induce effective phase separation between the PEDOT and PSS, which were initially bonded by ionic interaction, after transforming the PSS into PSSH to stabilize it as depicted in Figure 2b. This phase separation results in intensified π-π stacking interaction between polymers of the same kind, and thus the delocalization of electrons in the orbitals to form more continuous conductive channels between the PEDOT units to ultimately improve σ [57,58]. However, with 1.0 wt% HQ, a conductivity decrease to 672.3 S/cm was observed. This degradation with excess HQ was most probably caused by the limited HQ solubility in PEDOT:PSS solution which induced a weakened proton-donating effect that resulted in less PSSH formation [56].

The relationship between the HQ content and the performance of the fabricated HSCs was investigated by fabricating Si/HQ-PEDOT:PSS HSCs that contain various amounts of HQ, as shown in Figure 3. Figure 3b shows the current density vs. voltage (J-V) curve for the fabricated HSCs, and Table 1 summarizes the PV performance parameters. The PCEs of the HSCs (HQ-HSCs) containing HQ were higher than those of the pristine-PEDOT:PSS-based HSCs (Pristine-HSC) in all cases. This increase in the PCE is considered to be a result of the increase in σ of the HQ-PEDOT:PSS, as shown in a. However, as indicated in Table 1, we showed that the addition of HQ also dramatically improves the interfacial properties of Si/HQ-PEDOT:PSS because both the V_oc_ and shunt resistance (R_sh_) were measured to have improved. The improved interfacial properties are considered to be the consequence of the re-distribution within the coated layer of PEDOT and PSS, which have low and high W.F. characteristics, respectively [59]. As reported by many researchers, during the process of depositing the PEDOT:PSS layer using spin coating, a PEDOT-rich layer forms at the bottom, and a PSS-rich layer forms at the top [31,60]. However, as discovered in our previous study, the addition of HQ to pristine PEDOT:PSS promotes phase separation between PEDOT and PSS. As a result, PEDOT is more uniformly and continuously redistributed within the coated TF with the simultaneous formation of improved charge carrier transport paths. In other words, the addition of HQ also ensures that PSS is more uniformly distributed, rather than being present in excess near the top surface because of the formation of a PSS-rich top layer. This indicates that the addition of HQ increases the amount of PSS that is present at the Si/PEDOT:PSS interface [56]. The increased presence of PSS with its high W.F. in close proximity to the Si surface induces increased built-in potential (V_bi_) after the formation of a contact between Si and PEDOT:PSS. Consequently, for the HSCs, we can expect: (1) reduced carrier recombination loss at the surface owing to enhanced field effect passivation and (2) increased V_oc_ characteristics [40]. To confirm the re-distribution of PSS because of the presence of HQ to increase V_bi_, we applied the current-voltage diode equation [Equation (S1)] to the measured J-V curves for the dark condition to extract the V_bi_ values of the fabricated HSCs, as presented in Appendix A. As indicated by the table, all the HQ-PEDOT:PSS samples were observed to exhibit significantly increased V_bi_ values compared with those of the pristine-PEDOT:PSS. These results therefore enabled us to infer that the addition of HQ to pristine-PEDOT:PSS significantly increases not only the σ of PEDOT:PSS itself but also the interfacial properties of Si/PEDOT:PSS. As a result, the PCE of the HQ-HSCs would be expected to dramatically improve compared with that of the pristine-HSCs, as was confirmed in Table 1. As shown in the table, the PCE of the HSC with 0.67 wt% of HQ improved the most to 8.8% from that of pristine-HSC of 2.5%.

### 3.2. Comparison between HQ-PEDOT:PSS and Conventional DMSO-PEDOT:PSS

In this study, to confirm the advantages of using HQ-PEDOT:PSS to improve the HSC performance compared with conventional technologies, DMSO was added to PEDOT:PSS (DMSO-PEDOT:PSS). This is the most widely used method to improve the σ of PEDOT:PSS. Thus, a solution of DMSO-PEDOT:PSS was prepared and subsequently used to fabricate a Si/DMSO-PEDOT:PSS HSC (DMSO-HSC) to compare its performance with that of our HQ-HSCs. The measured J-V curves and detailed device parameters are presented in Figure 4 and Table 2, respectively. The results clearly show that replacing DMSO-PEDOT:PSS with HQ-PEDOT:PSS can increase the PCE by more than 10% from 7.7% to 8.8%. However, Table 2 reveals that the increase in the PCE resulting from HQ-PEDOT:PSS could mainly be attributed to not only the increase in J_sc_ (a result of the increased σ) but also to the increase in V_oc_ and R_sh_ which is consistent with the aforementioned HQ effect shown in Figure 3 and listed in Table 1.

We then compared the characteristics of the HQ- and DMSO-PEDOT:PSS TFs to clarify the reasons for the increased PCE with HQ-PEDOT:PSS compared to that of DMSO-PEDOT:PSS. First, the σ for each TF was measured and compared. As shown in Figure 5a, the σ of DMSO-PEDOT:PSS was significantly higher at 733.7 S/cm compared with that of pristine PEDOT:PSS (3.8 S/cm), which is similar to the level of σ in the previous report [33]. However, in the case of HQ-PEDOT:PSS, the σ of 936.8 S/cm was even higher than that of DMSO-PEDOT:PSS indicating that HQ increased the σ of PEDOT:PSS by an even greater amount than the conventional approach with DMSO.

As mentioned above, the increase in σ as a result of HQ is attributed to HQ acting as a proton-donating agent to PSS to promote the reduction of PSS to PSSH followed by the breakdown of the core-shell structure and ultimately by formation of more continuous linear-PEDOT domain within the TFs as depicted in Figure 5b [56]. However, on the basis of many previous studies, it is well known that, as a polar solvent, upon addition to pristine-PEDOT:PSS, DMSO would be placed between the PEDOT and PSS and would weaken the ionic interaction between them [61]. Therefore, the breakdown of the core-shell structure resulting from the presence of DMSO would also be expected to produce enhanced conductive-PEDOT continuity. We used AFM to analyze the surface morphology to compare HQ- and DMSO-PEDOT:PSS in terms of the extent to which the core-shell structure is broken down. In Figure 5c, the core-shell structure of the pristine PEDOT:PSS appears as islands of entangled bright regions [35]. However, the addition of DMSO (Figure 5d) and HQ (Figure 5e) significantly diminished the size of the islands and they became more uniformly and continuously distributed. However, in comparison, the degree of breakdown of the core-shell structure was more prominent for HQ-PEDOT:PSS. This is consistent with the results of the σ measurement shown in Figure 5a. Moreover, the comparison of the surface roughness (RMS) revealed the RMS of HQ-PEDOT:PSS to be 1.27, which is lower than the 1.77 of DMSO-PEDOT:PSS, confirming that HQ also provides a more uniform surface.

A more in-depth analysis of the increased PCE of HQ-HSC was carried out by additionally conducting XPS and UPS analyses to investigate the change in the PSS-to-PEDOT ratio and the effect on the W.F. of various PEDOT:PSS TFs. The XPS results are shown in Figure 6. As shown in Figure 6a, the S (2p) “curve” of pristine PEDOT:PSS comprises a spin-split doublet attributed to S(p_1/2_, _3/2_). The energy difference between the two peaks of the doublet is 1.18 eV, and the relative intensity between the doublets is known to be 1:2. In that case, the two peaks between 162 and 166 eV are spin-split doublets attributed to the sulfur atom in PEDOT, and the two peaks between 166 and 172 eV represent the sulfur atom in PSS [53,54]. Therefore, we calculated the area under the peaks of the binding energies using the S (2p) XPS results and compared the PSS-to-PEDOT ratio in DMSO- and HQ-PEDOT:PSS to analyze the change in the PSS-to-PEDOT ratio in each TF. The PSS-to-PEDOT ratio of the pristine-PEDOT:PSS (i.e., PH1000) used in our study is known to be 2.5 and we obtained a similar level of 2.45, as shown in Figure 6a. However, the addition of DMSO significantly decreases the value to 2.16, indicating that DMSO lowers the PSS ratio in the TF. This phenomenon was reported to commonly occur in many previous studies. This decrease in the PSS ratio occurs because the conventional approaches improve the σ of PEDOT:PSS by partly removing the insulator-like PSS during the deposition process. As a result, the ratio of conductive PEDOT in the coated TF increases [62,63]. This is because a polar solvent such as DMSO would provide a screening effect and would weaken the ionic interaction between PEDOT and PSS to promote the separation of PSS from PEDOT and induce the partial removal of PSS during spin coating [62,64].

However, as shown in Figure 6c, the PSS-to-PEDOT ratio of HQ-PEDOT:PSS remains at 2.42, which is only a minor change from the 2.45 of pristine PEDOT:PSS. This minor change strongly indicates that HQ enables more effective conductive channel formation in the PEDOT:PSS TF than DMSO even without the removal of PSS. This suggests that well-sustained high W.F. characteristics would be expected for PEDOT:PSS even with highly improved σ [39,65]. To confirm this, UPS measurements were recorded to compare the change in the W.F. of PEDOT:PSS TFs according to the degree of PSS removal, as shown in Figure 7. Based on the measured UPS results, a W.F. of 4.99 eV was measured for pristine-PEDOT:PSS, which is very close to its value of 5.0 eV [37,38]. However, the W.F. of DMSO-PEDOT:PSS was 4.87 eV, which is 0.12 eV lower than that of pristine PEDOT:PSS. In contrast, the W.F. of HQ-PEDOT:PSS was measured to be 4.98 eV, showing little change as expected. This high W.F. of HQ-PEDOT:PSS could be responsible for the higher PCE of our fabricated HSC (Table 2). Furthermore, the fill factor (FF) is expected to increase because of the high V_oc_ characteristics in addition to the more effective field effect passivation of the Si surface compared with that of DMSO-HSC.

### 3.3. n-Si/HQ-PEDOT:PSS Interface Engineering with Benzoquinone

Apart from the high σ of PEDOT:PSS, the properties of the Si/PEDOT:PSS interface are another crucial parameter to take into account when aiming to fabricate high-efficiency HSCs. Therefore, we also considered ways in which to improve the poor interface properties of Si/PEDOT:PSS. This made it necessary to consider two factors: (1) the formation of a passivation layer for excessive Si surface defect states and (2) improvement of the wettability of the PEDOT:PSS solution on the Si surface. Towards this purpose, we developed the technology to introduce benzoquinone (BQ) for Si surface passivation during HSC fabrication [66]. Figure 8 depicts the use of BQ to passivate the unsaturated-Si atoms on the Si surface. As shown in Figure 8a, BQ acts as an oxidation agent for MeOH when mixed with this solvent and transforms one of the carbonyl groups of BQ into a hydroxyl O-H group by acquiring a H^+^ from MeOH. BQ is then transformed into the semiquinone (SQ) molecule [66,67]. Upon immersion of the HF-cleaned Si substrate into the prepared BQ/MeOH passivation solution containing these SQ molecules, Si-SQ bonds are formed between unsaturated-Si atoms on the Si surface and the oxygen atom with an unpaired electron of SQ, as shown in Figure 8b. The formation of Si-SQ bonds basically provides a chemical passivation for the defect states on the Si surface. Moreover, as proven in our previous study, the Si-SQ bond also results in the formation of an intramolecular dipole moment owing to the difference in polarity between the O-H group and the oxygen atoms present at both ends of SQ. This arrangement induces upward band bending at the Si surface, as shown in Figure 8c [66]. This upward band-bending with SQ can lead to an imbalance in the carrier concentration between electrons and holes on the Si surface, eventually lowering the surface recombination velocity and thus providing effective field effect passivation for the Si surface.

Therefore, treatment of the Si surface with BQ can both lower the surface defect density and the carrier concentration on the surface, thereby decreasing the photo-generated carrier loss owing to the Shockley-Read-Hall recombination (R_SRH_), as expressed by Equation (1) below. As a result, the minority carrier lifetime (τ_eff_) within the devices could be expected to dramatically improve [68]:(1)$$R_{\mathrm{SRH}}=\left(n_{s}p_{s}-n_{i}{}^{2}\right)v_{\mathrm{th}}\times \int_{E_{v}}^{E_{c}}\frac{D\left(E_{t}\right)}{(n_{s}+n_{1})/\sigma_{p}\left(E_{t}\right)+\left(p_{s}+p_{1}\right)/\sigma_{n}\left(E_{t}\right)}{\mathrm{dE}}_{t}$$
where n_1_ and p_1_ refer to the density of electrons and holes in the bulk, n_s_ and p_s_ are the density of electrons and holes on the surface, $v_{\mathrm{th}}\;$ represents the thermal velocity, E_C_ and E_V_ refer to the conduction and valance band energies, D_it_ represents the density of interface states, and σ_n_ and σ_p_ are the energy-dependent capture cross sections for holes and electrons.

To compare the Si surface passivation efficiency of our BQ passivation method with that of the conventional native oxide approach, we measured the τ_eff_ for BQ and native oxide formed Si samples by fabricating the sandwich structures shown in Figure 9a. As a result, as shown in Figure 9b, BQ passivation yielded τ_eff_ of 133 μs, which is three times higher than that of the sample passivated by the native oxide. This confirmed that BQ treatment would considerably increase the passivation efficiency of a surface with excess Si defect states.

Additionally, it was also confirmed that the formation of Si-SQ bonds with BQ could improve the hydrophilicity of the Si surface. The results of the contact angle measurements are shown in Figure 10. These measurements were conducted to compare the wettability of the HQ-PEDOT:PSS solution between the BQ treated and native oxide formed Si surfaces. The results indicate that, the Si surface, on which the SQ layer is formed, has a smaller contact angle of 48.6° (Figure 10d) compared to 56.6° (Figure 10c) of the native oxide formed Si surface. The improved hydrophilicity is the result of the hydrophilic O-H group on the Si surface when the Si-SQ bond is formed, as depicted in Figure 8c [69,70].

To evaluate the increase in the PCE of the HQ-HSC treated with BQ, the device characteristics of HSCs with and without BQ treatment were evaluated, and the results are presented in Figure 11 and Table 3. The results in the table confirmed that BQ treatment notably increased the PCE of the fabricated HSC to 10.6%, which is more than 20% higher than that of the HSC with the native oxide. We consider this PCE improvement to be dominantly affected by: (1) the improved parasitic resistance (R_sh_ and R_s_) that originated from the formation of the hydrophilic SQ passivation layer and (2) higher V_oc_ resulting from the increased V_bi_ (i.e., $\Delta V_{\mathrm{bi}}^{\mathrm{SQ}}$) with the formation of a dipole moment with the SQ layer, as schematically illustrated in Figure 11b [67,71]. To confirm that the increase in the V_bi_ was a consequence of the BQ treatment, we extracted V_bi_ from the measured dark J-V characteristics for the fabricated HSCs as presented in Appendix A. The results reveal a higher V_bi_ for the BQ-treated HSC (BQ-HSC), 693 mV, compared to that of the native oxide formed HSC (SiO_x_-HSC), 661 mV. The results in Appendix A, obtained by calculation with the diode equation, show that V_bi_ increases during the formation of the SQ passivation layer. This indicates that SQ passivation would lower J_0_ by decreasing the R_SRH_ loss (i.e., by decreasing the ideality factor, n) as shown in Appendix A. Therefore, the BQ treatment introduced in this study can dramatically enhance the properties of the Si/PEDOT:PSS interface.

## 4. Conclusions

This report presents our novel technologies to improve the conductivity of PEDOT:PSS. Then, the properties of the Si/PEDOT:PSS interface were applied to fabricate a Si/PEDOT:PSS HSC with the ultimate objective of increasing the PCE. We successfully demonstrated that HQ-PEDOT:PSS can significantly improve the device performance compared to conventional approaches. Detailed analyses of the fabricated HSC and PEDOT:PSS TFs confirmed that the PCE improvement originated from the high σ and high W.F. of HQ-PEDOT:PSS. These improved properties were induced by the formation of effective charge transport paths within the conductive PEDOT and because the insulator-like PSS was not removed. In addition, we also showed that BQ treatment of the Si surface can produce an SQ layer on the surface after forming Si-SQ bonds. This layer not only acts as an effective passivation layer for Si surface defect states but also induces upward band-bending on the Si surface followed by a significant improvement of the properties of the Si/PEDOT:PSS interface. The use of these two approaches with HQ and BQ increased the PCE of the HSCs to 10.6%, an improvement of more than 20% compared to that achieved with the conventional approaches. We consider the HQ-PEDOT:PSS and BQ surface treatment technologies introduced in this study to provide an important foundation for developing low-cost and high-efficiency Si/PEDOT:PSS HSCs owing to their simple but highly effective processes whereby both the electrical properties of PEDOT:PSS and the interface properties of Si/PEDOT:PSS are improved.

## Figures and Tables

**Figure 1 polymers-14-00478-f001:**
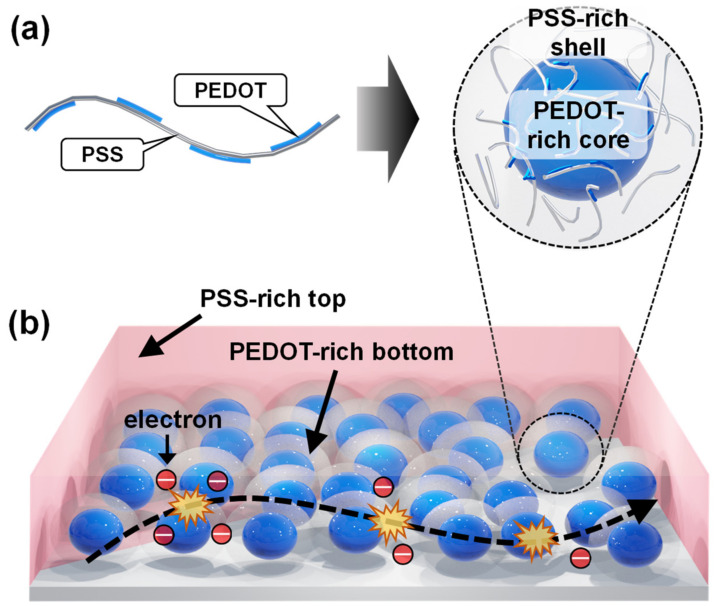
Schematic illustration of the (**a**) core-shell structure of PEDOT:PSS and (**b**) discontinuous distribution of conductive PEDOT in a coated pristine PEDOT:PSS TF.

**Figure 2 polymers-14-00478-f002:**
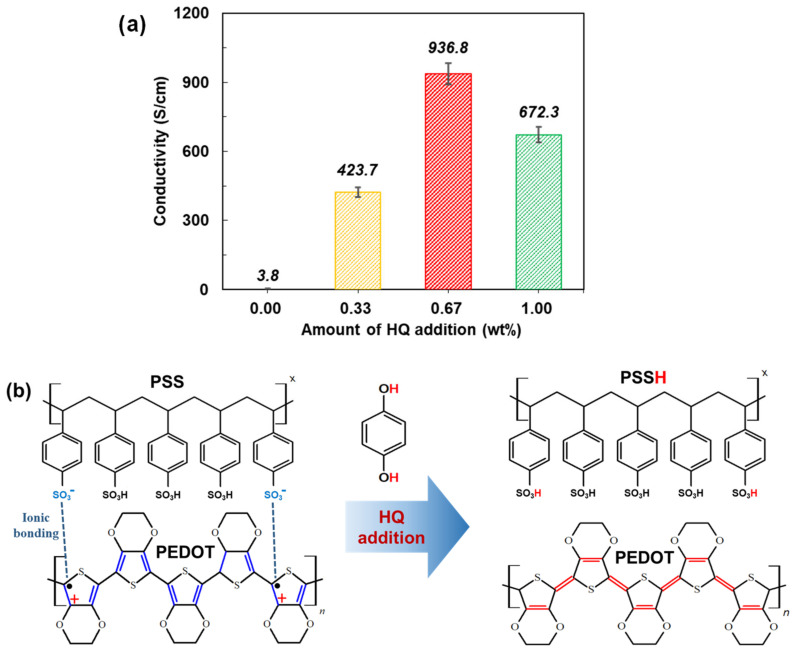
(**a**) Measured conductivity of PEDOT:PSS samples containing various amounts of HQ and (**b**) conformational change in the PEDOT:PSS structure after the addition of HQ.

**Figure 3 polymers-14-00478-f003:**
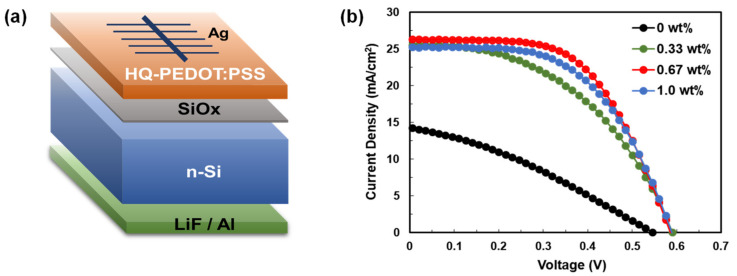
(**a**) Structure of the Si/HQ-PEDOT:PSS HSC and (**b**) current density vs. voltage characteristics of the various Si/HQ-PEDOT:PSS HSC samples with different HQ addition amounts.

**Figure 4 polymers-14-00478-f004:**
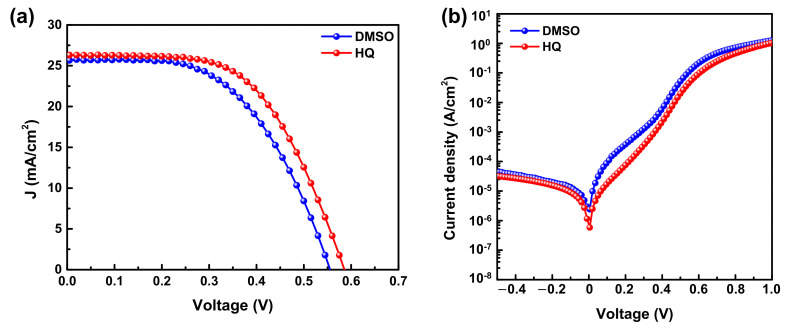
Current density vs. voltage curves of Si/PEDOT:PSS(DMSO, HQ) HSCs measured under (**a**) 100 mW/cm^2^ illumination (AM1.5) and (**b**) dark conditions.

**Figure 5 polymers-14-00478-f005:**
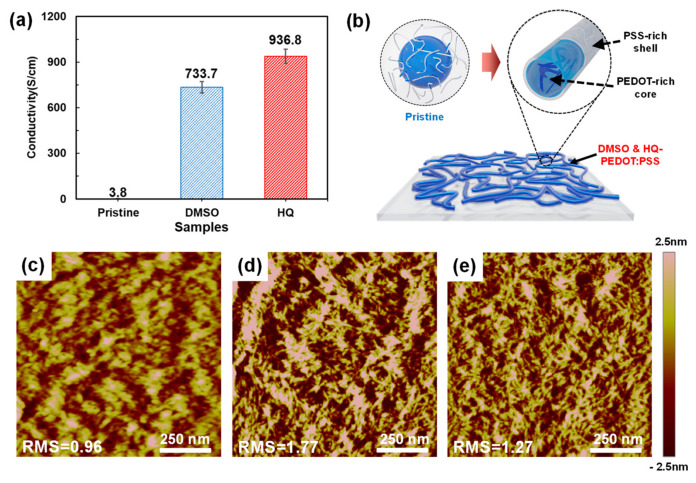
(**a**) Conductivity of various PEDOT:PSS TFs, and (**b**) schematic illustration of the HQ and DMSO effect on core-shell structure breakdown for continuous PEDOT conductive channel formation. AFM images of (**c**) pristine- (**d**) DMSO- and (**e**) HQ-PEDOT:PSS (scan size: 1 μm × 1 μm).

**Figure 6 polymers-14-00478-f006:**
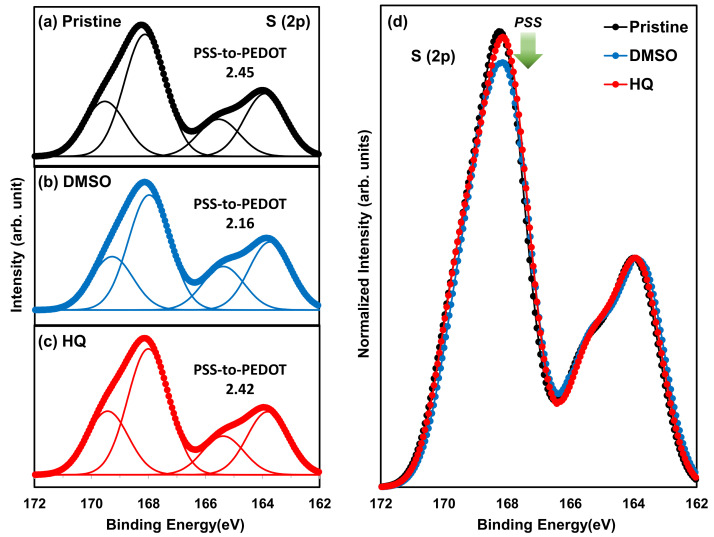
S (2p) XPS results of PEDOT:PSS: (**a**) pristine (**b**) DMSO (**c**) HQ and (**d**) relative comparison of peak intensities.

**Figure 7 polymers-14-00478-f007:**
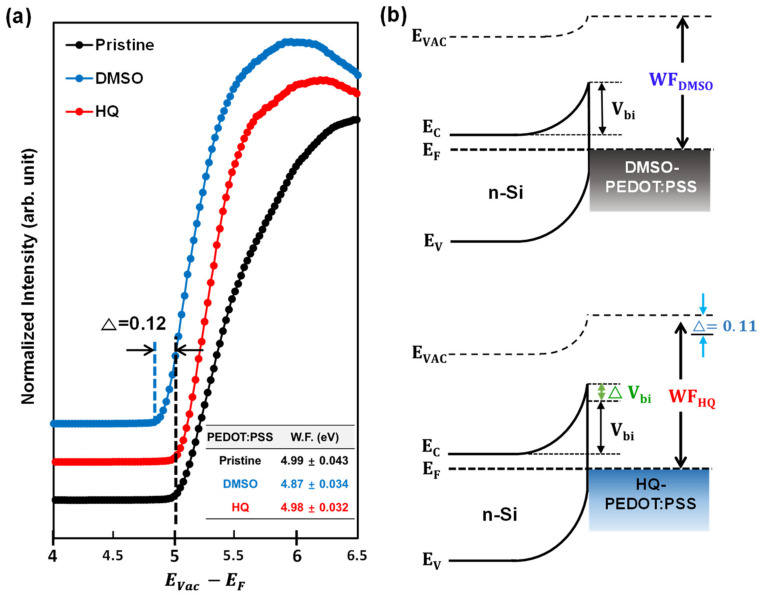
UPS results: (**a**) UPS secondary cut-off region and measured work function (W.F.) values and (**b**) energy diagram of cells based on the *n*-Si/PEDOT:PSS(DMSO, HQ) interface.

**Figure 8 polymers-14-00478-f008:**
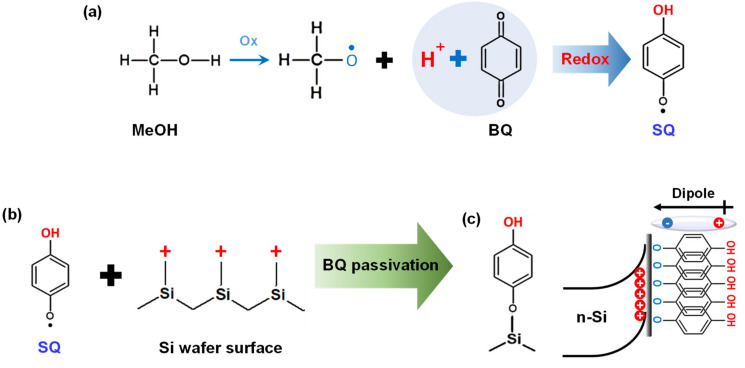
(**a**) Reduction reaction of BQ to form SQ, (**b**) schematic illustration of SQ passivation on Si surface, and (**c**) energy alignment of SQ molecules on *n*-Si surface.

**Figure 9 polymers-14-00478-f009:**
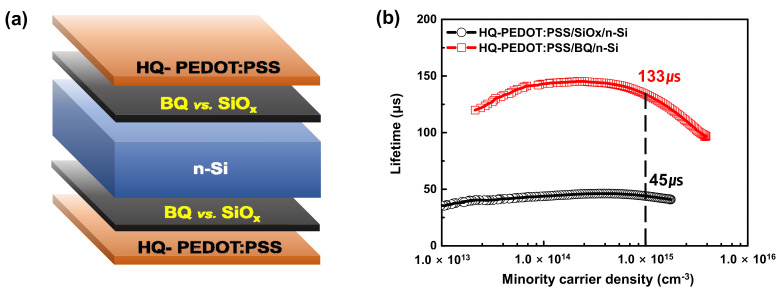
(**a**) The structure used to measure the lifetime and (**b**) carrier injection-dependent lifetime of HQ-PEDOT:PSS on silicon wafers with the different surface passivation layers.

**Figure 10 polymers-14-00478-f010:**
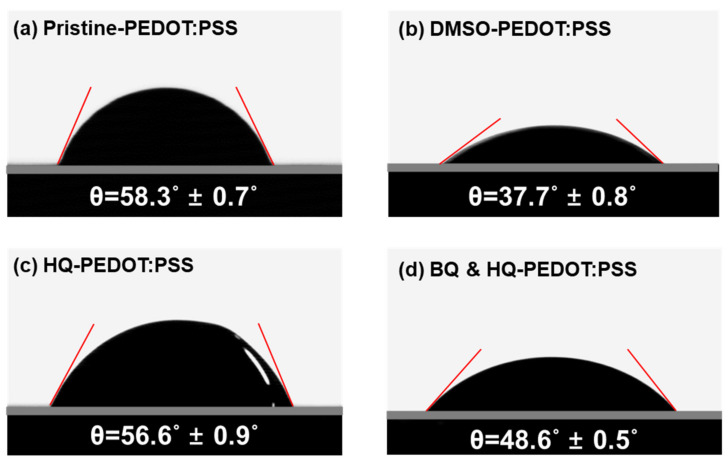
Measured contact angles of (**a**) pristine-, (**b**) DMSO-, (**c**) HQ-PEDOT:PSS on the native oxide formed Si surface, and (**d**) HQ-PEDOT:PSS on the BQ treated Si surface.

**Figure 11 polymers-14-00478-f011:**
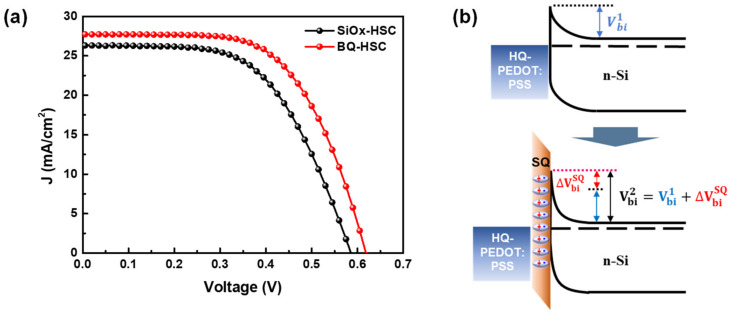
(**a**) Current density vs. voltage curves of SiO_x_-HSC and BQ-HSC measured under 100 mW/cm^2^ illumination (AM1.5) and (**b**) effect of interface dipole formation with BQ passivation of the Si surface.

**Table 1 polymers-14-00478-t001:** Photovoltaic parameters of Si/PEDOT:PSS HSCs containing various amounts of HQ.

Amount of HQ Addition	J_sc_ (mA/cm^2^)	V_oc_ (mV)	FF (%)	R_sh_ (Ω·cm^2^)	R_s_ (Ω·cm^2^)	PCE (%)
0 wt%	14.3	547	32.0	113.5	28.1	2.5 ± 0.93
0.33 wt%	25.6	590	47.1	504.8	6.8	7.1 ± 0.60
0.67 wt%	26.3	586	58.6	3680.9	6.1	8.8 ± 0.44
1.0 wt%	25.3	589	55.1	1569.1	5.8	8.2 ± 0.37

**Table 2 polymers-14-00478-t002:** Photovoltaic parameters of Si/PEDOT:PSS HSCs.

PEDOT:PSS	J_sc_ (mA/cm^2^)	V_oc_ (mV)	FF (%)	R_sh_ (Ω·cm^2^)	R_s_ (Ω·cm^2^)	PCE (%)
**DMSO**	25.7	556	53.8	1814.2	5.9	7.7 ± 0.86
**HQ**	26.3	586	58.6	3680.2	6.1	8.8 ± 0.44

**Table 3 polymers-14-00478-t003:** Photovoltaic parameters of SiO_x_-HSC and BQ-HCS.

Passivation	J_sc_ (mA/cm^2^)	V_oc_ (mV)	FF (%)	R_sh_ (Ω·cm^2^)	R_s_ (Ω·cm^2^)	PCE (%)
SiOx-HSC	26.3	586	58.6	3680.2	6.1	8.8 ± 1.36
BQ-HSC	27.7	618	61.8	4855.3	4.6	10.6 ± 1.59

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
