# Peer review of "Photovoltaic Device Application of a Hydroquinone-Modified Conductive Polymer and Dual-Functional Molecular Si Surface Passivation Technology"

_polymers, 2022, doi:10.3390/polym14030478_

Round 1

Reviewer 1 Report

My considerations in the manuscript “ High performance of Si/Organic Heterojunction Solar Cell with Enhanced Conductive Channel and Dual-functional Interface Engineering” are :  
1-    The title seems as the result. I feel it should be revised better.
2-    The abstract is well presented but also it starts with the result. Give the reader some background about the topic.
3-    Replace [Figure 1(b)] by (Figure 1(b))
4-    In figure 2 explain more why the conductivity reduced to 672.3 (S/cm) at 1 wt% of HQ.
5-    In this sentence relate to figure 4 “The results clearly show that replacing DMSO-PEDOT:PSS with HQ-PEDOT:PSS can increase the PCE by more than 10% from 7.7% to 8.8%” pleas justify why PCE increases?

Author Response

Dear reviewer-1,

First of all, I really appreciate your valuable and sharp comments to improve our manuscript. Here we provide author’s responses. Please find attached file for author responses.

Reviewer 2 Report

In the manuscript titled “ High performance of Si/Organic Heterojunction Solar Cell with Enhanced Conductive Channel and Dual-functional Interface Engineering”, Park et al. succeeded in improving the electrical conductivity of PEDOT:PSS to 937 S/cm through a simple process of adding hydroquinone (HQ) to the pristine PEDOT:PSS solution. The article is clearly described and the charts are beautiful. However, there are still some details need to be modified before further consideration.

Question 1:

What's new in this article compared to your previous reports,e. g. https://doi.org/10.1016/j.apsusc.2020.147176.

Question 2:

Some minor errors need to be corrected, (d) is missing in the title of Figure 5, and the AFM scale is inaccurate.

In the measurement of the contact angle in this work, the ellipse method is more suitable than the tangent method.

Author Response

Dear reviewer-2,

First of all, I really appreciate your valuable and sharp comments to improve our manuscript. Here we provide author’s responses. Please find attached file for author reponses.
